# Anxiety and Depression in the Initial Stage of the COVID-19 Outbreak in a Portuguese Sample: Exploratory Study

**DOI:** 10.3390/healthcare11050659

**Published:** 2023-02-23

**Authors:** Helena José, Cláudia Oliveira, Emília Costa, Filomena Matos, Eusébio Pacheco, Filipe Nave, Olga Valentim, Luís Sousa

**Affiliations:** 1Escola Superior de Saúde Atlântica, Fábrica da Pólvora, 2730-036 Barcarena, Portugal; 2The Health Sciences Research Unit: Nursing, Coimbra Nursing School, 3045-043 Coimbra, Portugal; 3Research in Education and Community Intervention, 4410-372 Vila Nova de Gaia, Portugal; 4Escola Superior de Saúde Jean Piaget Algarve, 8300-025 Silves, Portugal; 5Nursing Departamento, Escola Superior de Saúde da Universidade do Algarve, 8005-139 Faro, Portugal; 6Nursing School of Lisbon (ESEL), 1600-096 Lisboa, Portugal; 7Nursing Research, Innovation and Development Centre of Lisbon (CIDNUR), Av. Prof. Egas Moniz, 1600-096 Lisboa, Portugal; 8Center for Health Technology and Services (CINTESIS@RISE), Research Group Innovation & Development in Nursing (NursID), 4200-450 Porto, Portugal; 9Comprehensive Health Research Centre, 7000-811 Évora, Portugal

**Keywords:** anxiety, depression, COVID-19, social isolation, mental health

## Abstract

Background: In previous studies, it was found that the confinement to which the population was subjected during the quarantine of the COVID-19 pandemic increased the risk of anxiety and depression. Objective: to analyze the levels of anxiety and depression symptoms in Portugal residents during the quarantine of the COVID-19 pandemic. Methods: This is a descriptive, transversal, and exploratory study of non-probabilistic sampling. Data collection was carried out between 6th and 31st of May 2020. Sociodemographic and health questionnaires PHQ-9 and GAD-7 were used. Results: The sample consisted of 920 individuals. The prevalence for depressive symptoms (PHQ-9 ≥ 5) was 68.2% and (PHQ-9 ≥ 10) was 34.8%, and for anxiety symptoms (GAD-7 ≥ 5) was 60.4% and (GAD-7 ≥ 10) was 20%. Depressive symptoms were moderately severe for 8.9% of the individuals, and 4.8% presented severe depression. Regarding the generalized anxiety disorder, we found that 11.6% of individuals present moderate symptoms, and 8.4% severe anxiety symptoms. Conclusions: The prevalence of depressive and anxiety symptoms were substantially higher than those previously found for the Portuguese population and when compared with other countries during the pandemic. Younger individuals, female, with chronic illness and medicated, were more vulnerable to depressive and anxious symptoms. In contrast, participants who maintained frequent levels of physical activity during confinement had their mental health protected.

## 1. Introduction

Since it appeared in China in late 2019 [1], SARS-CoV-2 has infected more than one hundred million people, with more than one million people in Portugal (October 2021). After the pandemic was declared, most governments, including the Portuguese, imposed confinement as a protective measure. This procedure brought abrupt and disruptive changes in people’s ways of living, working, and socializing. On 16 March, quarantine was imposed in Portugal, with school closures and lockdowns on 22 March for 45 days, compelling citizens to staying-at-home duty to control virus dissemination. Several studies show the negative impact of COVID-19 on population mental health [2,3,4,5] because they fear being infected and due to the need for physical and social isolation. Confinement restraint conviviality subtracted affection and changed people’s relation patterns. These aspects affect mental health. In fact, several studies shown an increased prevalence of depression symptoms and anxiety [6,7,8,9,10].

A study developed in China found that 54% of the respondents present psychological distress related to the disease’s outbreak. Specifically, 17% reported moderate to severe depression symptoms, 29% anxiety and 8% stress [11]. Another Chinese study showed a prevalence of severe depression symptoms (48.11%) and severe anxiety (53.46%) among those infected; similar data were found in people with high risk of infection [12]. These consequences were recognized by the National Health Committee of China, which called for an intervention through several mental health associations to provide support to the general population [11] as well as to health professionals [13]. This situation also occurred in Portugal and the Portuguese Order of Nurses created a mental health support line, as well as other professional orders. In a Spanish study 18.7% revealed depression and 21.6% anxiety [4]. In France, research with university students showed that 43% had depression and 39.19% anxiety, with 20.7% having a high level of anxiety [14].

Undoubtedly, the COVID-19 pandemic and the associated measures to protect populations, such as confinement, have a strong, negative impact on the population’s wellbeing. The pandemic crisis challenges equity in access to health care. This may arise due to the scarcity of resources, which hinder both the entry into the care system itself and the access to life-sustaining resources. This situation is particularly worrying for people with chronic diseases, due to the associated comorbidities, but also because they are the ones who recurrently need health care [15]. These aspects contribute to increased risk for these persons, who are two to three times more likely to have mental health problems, such as anxiety [16].

Therefore, and despite the growing evidence that COVID-19 pandemic seriously affects people’s mental health, it is still necessary to clarify this sphere and, specifically, to elucidate how confinement reverberates on Portuguese mental health, to establish protective public policies. Thus, this study aims to analyze the levels of anxiety and depression symptoms in Portugal residents during the confinement of the COVID-19 pandemic.

## 2. Materials and Methods

### 2.1. Study Design and Participants

It is a descriptive, cross-sectional, and exploratory study of non-probabilistic sampling. Data collection was carried out using the Google Forms platform sent via social networks (Facebook, Instagram, and WhatsApp) between 6 and 31 May 2020.

Sample size was calculated a priori using G × Power 3.1.9.7 software. In the case of depressive and anxiety symptoms, the effect sizes were based on the minimal clinically important differences (MCID) for depressive symptoms (Cohen d  =  0.24) [17] and anxiety (Cohen d  =  0.29) [18]. In order to use ANOVA with two groups, effect size of Cohen d  =  0.24, statistical power (1-B) of 90%, and Alpha of 5%, the minimum sample required was 186 participants.

The initial sample consisted of 929 participants. Participants with more than 20% of missing data were excluded (9 participants). Finally, included in the sample participants (*n* = 920) were those with the following criteria: (a) 18 years old and more; (b) Portugal resident; (c) be confined for at least 15 days; (d) be able to read and understand Portuguese; (e) voluntarily accept to participate in the study after observing the Free and Informed Consent Term (ICF). Given the characteristics of the data collection tool, we considered as exclusion criteria low digital competence, the impossibility of accessing the internet or not having social networks and not answering at least 80% of the total questions on the form.

### 2.2. Measures

-Sociodemographic and health questionnaire: age; nationality; sex; marital status; residence region; profession; proximity to a family member or friend who has or has had the Coronavirus; confinement duration; presence of chronic disease (these data allowed us to assess whether, before the start of the pandemic, individuals already had some type of chronic pathology, namely mental and/or psychiatric); use of medications; regular physical activity (minimum 3 times a week); health facilities resort; doctor or psychologist appointment; alcoholic habits. How satisfied are you with your health? With the following answer possibilities: Very unsatisfied, Unsatisfied, Neither dissatisfied nor satisfied and Satisfied. How satisfied are you with your family economic income? With the following answer possibilities: Very unsatisfied, Unsatisfied Neither dissatisfied nor satisfied, Satisfied and Very Satisfied. With regard to spirituality, during quarantine a person is considered: Unspiritual, With some spirituality, Not too little, not too spiritual, Spiritual and Very spiritual.-Patient Health Questionnare-9 (PHQ-9), developed by Kroenke, Spitzer and Williams (2001) [19], validated for the Portuguese population by Monteiro et al. (2013; 2019) [20,21], is a self-report questionnaire asking about the nine items of major depression disorder (Diagnostic and Statistical Manual of Mental Disorders–DSM-IV), during the last two weeks. For each item, responses are rated on a 4-point Likert scale, ranging from 0 (“Not at all”) to 3 (“Almost every day”). Higher scores represent high levels of depressive symptoms, which can vary from 0–27 [22]. We considered the recommendations of Kroenke et al. (2001): minimal 0–4, mild 5–9, moderate 10–14, moderately severe 15–19, and severe 20–27 [19]. Global scale Cronbach’s α is 0.88 [23] and in the present study 0.89.-General Anxiety Disorder (GAD-7), developed by Spitzer, Kroenke, Williams and Löwe (2006) [22], translated and validated for the Portuguese population by Sousa et al. (2015) [24]. GAD-7 is a one-dimensional instrument, composed by 7 items assessing the presence of anxiety symptoms in the last 14 days. Uses a Likert scale ranging from 0–3 (0 = nothing, 1 = several days, 2 = more than half the days and 3 = almost every day). The sum of the items allows a total anxiety score, ranging from 0–21. Severity is determined by the cutoff scores; 0–4 normal, 5–9 mild symptoms, 10–14 moderate symptoms and 15–21 severe symptoms. The global scale Cronbach’s α is 0.93 [25] and in the present study 0.92.

### 2.3. Statistical Analysis

Statistical analysis SPSS^®^ software, version 26, was used for data analysis. In the statistical data treatment, in addition to the descriptive and exploratory analysis, absolute (*n*) and relative (%) frequencies were used for qualitative variables, as well as measures of central tendency: mean (M), standard deviation (SD), minimum (min.), maximum (max.) and range. Means were compared between groups, using the Student’s Test and the Simple Analyze of Variance (ANOVA) with Bonferroni’s correction. To identify associations between two categorical variables, Pearson’s chi-square tests were used. The results of the present study are considered statistically significant for a level of significance below 5% [26]. The odds ratios (ORs) of anxiety and depression symptoms were also calculated by step-by-step multiple logistic regression model from which odds ratios for each explanatory variable with the corresponding 95% confidence interval (CI) and *p*-value were presented. We used cut-off point ≥10 to GAD-7 and PHQ-9 scales to create the dichotomous variables.

## 3. Results

### 3.1. Sociodemographic and Health Characteristics

The sample consisted of 920 Portuguese residents, 79.2% in the southern region of the country, 72.4% were women. Average age was 42.6 years (SD = 13.2; range = 18–72), 42.0% were married and 30.8% single. A total of 41% were health professionals and from that, 74.3% were nurses.

Regarding the period of quarantine, a mean value of 40,1 days was found (SD = 23.8; range = 15–125). In total, 37.9% of the participants were teleworking, 17.6% kept working as usual, and 14.9% worked with reduced/adapted schedule. Most of the participants (85.5%) reported not having any family or friends with COVID-19 in this period.

Concerning the clinical history, 74.9% of the individuals reported not having chronic disease and 61.8% reported not using any type of medicine daily. A total of 2.7% (SD = 2.4) refer the use of medicines and 2.6 (SD = 2.3) the use of prescription medicines. In total, 85.4% did not resort to a health service, neither a doctor nor a psychologist (88.7%).

Analyzing this period, it was found that 56.3% of the respondents did not participate in regular physical activity. In addition, 55.0% did not consume alcoholic beverages, but 16.6% had increased their alcohol consumption.

### 3.2. Depressive Symptoms and Generalized Anxiety Disorder

Regarding the main variables, a mean of 8.13 was found for the depressive symptoms (SD of 5.89); concerning the generalized anxiety disorder, the mean found was 6.24, with SD of 4.95 (Table 1 and Table 2).

In the sample studied, 68.2% (PHQ-9 ≥ 5 score) of the participants had depressive symptoms. These symptoms were moderately severe for 8,9%, and 4,8% presented severe depression symptoms (Table 1). For the GAD-7 ≥ 5 score, 60,4% referred to anxiety symptoms, 11,6% moderate symptoms, and 8,4% severe anxiety symptoms (Table 2).

After data analysis, relationships between different variables were clarified. Table 3 shows the statistically significant results.

Regarding PHQ-9, statistically significant differences were found for several variables, videlicet, for age (F = 4.265; *p* = 0.014), the 37-years-old or less group had the highest mean (8.8); for sex (t = 5.276, *p* < 0.0001), women had the higher average value (8.7); also, for the health professional type (F = 3.726; *p* = 0.012), the Diagnostic and Therapeutic Technicians (DTT) had the highest mean value (11); chronic diseases patients (t = −2.424, *p* = 0.016) had a mean value of 8.9; those who daily used medicines and prescribed drugs (t = −4.766, *p* < 0.0001 and t = −4.802, *p* < 0.0001) showed mean values of 9.4 and 9.3, respectively; in physical activity, we found higher mean values (8.9; t = 4.873, *p* < 0.0001) in individuals who did not practice it regularly; the respondents who reported having resorted to health facilities during the quarantine also have higher mean values (9.3, t = −2.510, *p* = 0.012) (Table 3).

For the GAD-7, significant differences were found for several variables, namely age (F = 9.271, *p* < 0.0001), 37-years-old or less group showed higher mean values (6.8); sex (t = 4.633, *p* < 0.0001), women are most vulnerable with a mean value of 6.7; type of health professional (F = 3.233, *p* = 0.022), observing a higher score in DTT (8.1); daily use of medicines and prescribed drugs (t = −3.19, *p* = 0.001 and t = −3.338, *p* = 0.001, respectively), with a mean value for both variables of 6.9; for physical activity during the quarantine, we found that participants who reported not having practiced it regularly have high mean values (6.8; t = 3.699, *p* < 0.0001); individuals who resorted health facilities presented a mean value of 7.1 (t = −2.372, *p* = 0.018) (Table 3).

Through multivariate analysis, predictors of depressive and anxiety symptoms were determined. As predictors of depressive symptoms, age, sex, satisfaction with health and economic income, as well as spirituality, were found. Depressive symptoms were more likely to be present in younger people, women, people who were dissatisfied with their health and economic performance and reported being unspiritual.

Predictors of generalized anxiety symptoms in this sample were age, sex, satisfaction with health and economic income. That is, anxiety symptoms were more likely to occur in younger people, women and in people who are dissatisfied with their health and economic performance and unspiritual (Table 4).

## 4. Discussion

This investigation aimed to analyze the levels of generalized anxiety disorder and depressive symptoms during COVID-19 confinement in Portuguese residents. Additional objectives meant to explain how sociodemographic and health variables were associated with these mental health indicators.

The global dissemination of COVID-19 had an important impact on individuals’ lives. It threatened individuals’ physical health, installed fear of contagion and possible transmission to more vulnerable family members and gradually affected other dimensions of individual and collective health. In recent history there is no record of anything comparable to the need for prophylactic social isolation, extended to millions of people and without a definite end, a fact that places mental health at risk [27]. Although, with different outlines regarding the size of the population covered by the quarantine and its duration, other investigations have shown the negative impact of prophylactic isolation on mental wellbeing [28,29].

Portugal has an annual prevalence of mental illnesses of 22.9% [25,30], among which anxiety and mood disorders stand out [31].

These values are substantially higher than those found in other European countries, a fact that has not yet been fully explained and which may eventually be associated with greater exposure to factors that cause vulnerability or less exposure to protective factors [25]. It will thus be expected that, starting from a more unfavorable situation, the risk to mental health is greater when facing a challenge, such as the current pandemic.

### 4.1. General Anxiety Disorder (GAD-7)

We accessed mental health by measuring depressive symptoms and generalized anxiety disorder. Regarding this last construct, we found that 60.4% of the respondents had symptoms of anxiety (GAD-7 ≥ 5), 11.6% moderate symptoms and 8.4% severe anxious symptoms. Thus, we found a prevalence of 20% (GAD-7 ≥ 10) of relevant anxiety symptoms, a higher value than the one found in the study (using the Composite International Diagnostic Interview) that compared the prevalence of anxiety disorders in Portugal, some countries in Europe and the USA, where Portugal presented the second highest value in the group (16.5%) [30]. The values found in the present study are also substantially higher than those found in Germany, also in a pre-COVID situation, identified only 5.1% of individuals with GAD-7 ≥ 10 [32]. These differences underline the important impact that the pandemic and the subsequent global crisis has had on the mental health of individuals.

Contrasting our findings with research already carried out in this pandemic context, we find that Portuguese values (GAD-7 ≥ 10) are also higher than those evidenced in other studies, such as the investigations in Hong Kong with 14% [33], and in Austria with 19% [34]. In a German study [35], a value of 44.9% (GAD ≥ 5) was found, also lower than what we found (60.4%) for the same cutoff point. It should also be noted that the values we found are close to the values found in the INSA study (with a larger number of subjects and developed for a longer time than the present study) which show 27% of moderate to severe anxiety symptoms in the general population during the pandemic situation [36]. These data indicate that the vulnerabilities previously demonstrated for the Portuguese population put them, at the outset, in a more adverse situation to face the current challenges, showing higher values of mental health fragility. The prevalence of anxiety in low- and middle-income countries (35.1%; 95%CI: 29.5% to 41.0%) was similar to that of high-income countries (34.7%; 95%CI: 29.6 % to 40.1%), that is, one in three people felt anxiety during the pandemic [37].

### 4.2. Depressive Symptoms (PHQ-9)

Depressive symptoms were found in 68.2% of the participants: 8.9% moderately severe; 4.8% severe. Thus, we can affirm that in the studied sample the prevalence of relevant depressive symptoms was 34.8% (PHQ-9 ≥ 10) and 13.7% (PHQ-9 ≥ 15). These values, in line with the findings related to the generalized anxiety construct, are also higher than the 7.9% prevalence of depressive disorders presented by DGS [30].

For this variable, the values found in our sample are also higher than other studies in the same context, namely the investigations carried out in Austria [34], in France [38], in China [11] and in Cyprus [39]. It should be noted that, when compared with the data from Almeida et al., (2020), the values we found for a PHQ-9 ≥ 10 (35.8%) are higher than the data found by these authors, whether in the general population (26.4%) or among health professionals (28.4%). Once again, these findings documented the risky situation that the Portuguese population is facing due to the impact of the pandemic on their lives. The overall prevalence of depression was 28.18% (95% CI: 23.81–32.54) [40]. In another meta-analysis, a prevalence of depression of 23.2 was found [41].

### 4.3. Socioeconomic and Health Variables Related to Anxiety and Depression Symptoms

The present study allowed us to underline the impact of the pandemic on mental health, and made it possible to verify that certain individuals’ variables were significantly associated with it. We confirmed that the youngest participants, the women, the ones who consume prescription drugs daily, the ones who resorted health services during quarantine and the ones who did not practice physical activity on a regular basis, presented simultaneously higher values of anxious symptoms and depressive symptoms, therefore appearing to present greater vulnerability to these problems.

Although it may seem unexpected that younger individuals have higher levels of anxious and depressive symptoms, this finding has also been found in other studies [37,39,40,42]. It seems consensual that young adults are more exposed to media information and that can be a factor of greater stress, they may also be more sensitive to issues arising from social isolation and fear more for their future as they belong to the most active fringe of the population, whether students or already workers. This aspect may also be related to issues of maturity and resilience in the face of adversity, skills that are being developed later in life.

The levels of anxious and depressive symptoms in this study are significantly associated with sex, showing important differences in the distribution of these two variables, presenting women higher values than men. Women are more vulnerable to stress-generating stimuli and their impact on mental health, as several previous investigations have documented [11,39,43,44,45,46].

Notwithstanding, no significant differences were found in the levels of anxiety or depression due to the fact that the respondents were health professionals or not, a situation also verified in another study [47], we found that among health professionals, DTT showed significant differences, with higher scores for anxiety and depression. In a previous study it was found that females and nurses had more symptoms of depression than males and doctors [41]. This finding contradicts data from other studies [48,49], in which the levels of anxiety and depressive disorders were more evident in nurses or doctors because they are the professionals in more direct contact with patients and their families. In this situation, we can conjecture that nurses, being more frequently exposed to situations of higher levels of stress, will have more expertise in adopting appropriate coping strategies and also higher levels of resilience, which will function as a protective process to the current emotional overload [50].

The presence of chronic disease and the daily intake of medication discriminated aspects related to individuals’ mental health. The persistent information in media about the increased risk associated with COVID-19 for people with comorbidities has certainly contributed to the intensification of psychological suffering, developing in these individuals a perception of insecurity and fear. This may have raised the level of anxious and depressive symptoms or even their worsening, when pre-existing, these data are validated by several studies developed in this context [51,52].

The participants who resorted to a health facility during confinement, reported higher levels of both anxiety and depressive symptoms which may be related to the perception of risk inherent to the contact with a health service. The data may also be related to the necessity for certain individuals with specific health needs (namely chronic illness) to use these resources more frequently and to combine, at the same time, several factors of vulnerability to psychological distress. Several studies have found similar data in previous pandemics [53] and the current situation [38].

In our study, we found that the most physically active individuals during this period had lower levels of depressive and anxious symptoms. In fact, the beneficial effect of physical activity on the overall health of individuals and particularly on their mental health, well-being and cardiovascular health is consensual, in the pre-pandemic period [54,55] and during confinement due to COVID-19 [56,57]. We also want to underline that, although considering the characteristics of this study, it is not possible to speak of a cause-and-effect relationship between physical activity, depression and anxiety, a significant association between the practice of physical activity and better mental health indicators is clearly demonstrated. Furthermore, frequent physical activity seems to have had a protective effect during confinement. The findings from systematic reviews show that there is evidence that physical activity can improve immune functioning in situations of restriction (such as confinement), especially in the most vulnerable populations reviews [58].

As in our study, other studies point to similar results, in which people at high risk of mental problems are women, young people, single people and those with a low educational level. Intervention in anxiety and depression in a pandemic context includes pharmacological treatment, psychological therapy, and physiotherapy [59].

Some limitations may affect the interpretation of the results of this study, namely the large number of health professionals in the sample, as well as the data collection technique used. The fact that we used personal and professional social networks may have created this bias, which led to a sample with an important representation of health professionals. Other limitations of the present study are, we present the way which data collection was carried out, as we are aware that not all people residing in Portugal have access to e-mail, internet, or WhatsApp. Finally, another limitation is the fact that 36% of health professionals continued to work, which means that they were not in confinement, which can represent a bias. However, the situation can be seen from another perspective, because even the health professionals who continued to work were transversally prevented from maintaining their family relationships, which, we think, will not be far removed from what the confined people experienced.

## 5. Conclusions

The research showed that during the first confinement of SARS-CoV-2, the mental health of the Portuguese participants in the study was at risk. This fact is reflected in a prevalence for depressive symptoms and for anxious symptomatology showing that our values are substantially higher than those found for the Portuguese population prior to the pandemic and in research carried out during the current situation.

We were able to verify that depression and anxiety are more likely to be present in younger people, women, people who were dissatisfied with their health and economic performance and reported being unspiritual. In contrast, participants who maintained more frequent levels of physical activity during the confinement apparently had their mental health protected.

These findings can be used to develop tailored intervention strategies and psychological support for fringes of vulnerable population, aiming the maintenance or upsurge of people’s mental health and preventing or minimizing emotional suffering in civilizational crises, such as the one we are experiencing. This intervention process must be a health priority considering that the implications to mental health will rest (surely) longer than the pandemic itself and its psychosocial impact may be incalculable.

Further studies will be necessary to verify the consistency of these results since the pandemic brought the development of new habits, new risks, new vulnerabilities that tests people’s resilience and capacity to mobilize their internal and external resources to face the challenges that emerge.

## Figures and Tables

**Table 1 healthcare-11-00659-t001:** Depressive symptoms distribution.

Variable	*n*	Mean (Min./Max.)	St. Dev.	Normal (0–4) *n* %	Mild (5–9) *n* %	Moderate (10–14) *n* %	Moderate Severe (15–19) *n* %	Severe Depression (20–27) *n* %
Depressive Symptoms (PHQ–9)	920	81,283 (0–27)	589,211	293 31.8%	298 32.4%	203 22.1%	82 8.9%	44 4.8%

**Table 2 healthcare-11-00659-t002:** Generalized anxiety disorder distribution.

Variable	*n*	Mean (Min./Max.)	St. Dev.	Normal (0–4) *n* %	Mild (5–9) *n* %	Moderate (10–14) *n* %	Severe (15–21) *n* %
Generalized Anxiety Disorder (GAD–7)	920	62,435 (0–21)	495,304	364 39.6%	372 40.4%	107 11.6%	77 8.4%

**Table 3 healthcare-11-00659-t003:** Depressive symptoms and generalized anxiety levels during the quarantine, according the Sociodemographic and Health variables (*n* = 920).

Variables	Categories	Total *n* %	Patient Health Questionnaire (PHQ)		Test (*p*-Value)	Generalized Anxiety Disorder Scale (GAD7)		Test (*p*-Value)
			Mean	SD		Mean	SD	
Age	<=37	334 (36.3%)	8.8	5.4	* F = 4.265 (*p* = 0.014)	6.8	4.9	* F = 9.271 (*p* < 0.0001)
	38–49	294 (32.0%)	8.1	6.3		6.5	5.4	
	>=50	292 (31.7%)	7.4	5.9		5.2	4.4	
Sex	Female	655 (72.4%)	8.7	5.9	** t = 5.276 (*p* < 0.0001)	6.7	5.0	** t = 4.633 (*p* < 0.0001)
	Male	253 (27.6%)	6.5	5.4		5.0	4.4	
Health Professional Type	Nurse	280 (74.3%)	8.0	5.7	* F = 3.726 (*p* = 0.012)	6.2	4.8	* F = 3.233 (*p* = 0.022)
	Doctor	32 (8.5%)	7.5	5.8		5.1	4.2	
	Therapeutic and Diagnosis Technician	30 (8.0%)	11.0	5.9		8.1	5.3	
	Other	35 (9.3%)	6.4	5.5		4.7	5.0	
Chronic Disease	No	689 (74.9%)	7.8	5.6	** t = −2.424 (*p* = 0.016)			
	Yes	231 (25.1%)	8.9	6.5				
Use of Medicines	No	569 (61.8%)	7.4	5.3	** t = −4.766 (*p* < 0.0001)	5.8	4.7	** t = −3.191 (*p* = 0.001)
	Yes	351 (38.2%)	9.4	6.6		6.9	5.3	
Prescribed Medicines	No	561 (61.0%)	7.4	5.3	** t = −4.802 (*p* < 0.0001)	5.8	4.6	** t = −3.338 (*p* = 0.001)
	Yes	359 (39.0%)	9.3	6.6		6.9	5.4	
Regular Physical Activity	No	518 (56.3%)	8.9	6.1	** t = 4.873 (*p* < 0.0001)	6.8	5.0	** t = 3.699 (*p* < 0.0001)
	yes	402 (43.7%)	7.1	5.4		5.6	4.8	
Resort to Health Unit	No	786 (85.4%)	7.9	5.8	** t = −2.510 (*p* = 0.012)	6.1	4.9	** t = −2.372 (*p* = 0.018)
	Yes	134 (14.6%)	9.3	6.4		7.1	5.4	

* ANOVA. ** Student-t test.

**Table 4 healthcare-11-00659-t004:** Logistic regression analysis of depressive and anxiety symptoms and their predictors.

Variables	Categories	Patient Health Questionnaire-9 (PHQ-9) ≥ 10	General Anxiety Disorder (GAD-7) ≥ 10
		Odds Ratio (CI95%)	Odds Ratio (CI95%)
Intercept		0.021 (0.007−0.061)	0.005 (0.001–0.024)
Age	<=37	1.397 (0.964–2.024)	2.040 (1.304–3.190)
	38–49	1.011 (0.688–1.486)	1.544 (0.972–2.454)
	>=50	1	1
Sex	Woman	2.098 (1.461−3.014)	1.901 (1.223–2.954)
	Man	1	1
How satisfied are you with your health?	Very unsatisfied	3.828 (1.512–9.692)	8.009 (2.857–22.457)
	Unsatisfied	7.015 (3.527–13.951)	6.800 (3.109–14.872)
	Neither dissatisfied nor satisfied	4.318 (2.676–6.968)	4.347 (2.296–8.231)
	Satisfied	1.487 (0.934–2.366)	2.211 (1.169–4.183)
	Very Satisfied	1	1
How satisfied are you with your family economic income?	Very unsatisfied	4.093 (1.685–9.940)	3.719 (1.254–11.031)
	Unsatisfied	3.277 (1.424–7.541)	2.126 (0.741–6.101)
	Neither dissatisfied nor satisfied	1.618 (0.745–3.514)	1.222 (0.445–3.352)
	Satisfied	1.122 (0.506–2.486)	1.199 (0.429–3.350)
	Very Satisfied	1	1
With regard to spirituality, during quarantine a person is considered:	Unspiritual	4.445 (1.984–9.962)	6.578 (2.132–20.290)
	With some spirituality	3.667 (1.624–8.283)	4.237 (1.345–13.350)
	Not too little, not too spiritual	3.771 (1.802–7.891)	4.702 (1.603–13.796)
	Spiritual	3.809 (1.776–8.167)	4.237 (1.406—12.763)
	Very spiritual	1	1

## Data Availability

The data used during this study are available from the corresponding author, under request by email.

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
