# Peer review of "Anxiety and Depression in the Initial Stage of the COVID-19 Outbreak in a Portuguese Sample: Exploratory Study"

_healthcare, 2023, doi:10.3390/healthcare11050659_

Round 1
Reviewer 1 Report
This study determined the levels of anxiety and depression symptoms in Portugal residents during the confinement of the COVID-19 pandemic. The authors described the study well. However, a few comments need attention from the authors to improve this manuscript.
1. The statement for sample size was not clear. In your method, you did mention that the sample participants were 920 (Line 77). There were 9 participants excluded due to missing data, more than 20% (Line 81). However, in your result section, your analysis included 920 participants – has this excluded the 9 participants with missing data? How many participants were used in the analysis, and what was the sample size calculated for this study?
2. Please include in the footnote of Table 3 to differentiate which variable used Student-t and ANOVA test. Why do you need both when ANOVA can cater to two groups and above? Suggest using ANOVA.
3. In Table 3, only the health profession was analyzed. What about other professions since the data involved Portuguese residents, not particularly from the health sector only?
4. Categorical variables depression and anxiety in the original scale were 5 subcategories (Line 98) and 4 subcategories (Line 107-108), respectively. Why were they categorized into 3 and 2 subcategories by Visual binning in SPSS (Line 124-125)? Provide the justification and support for such. And what did the new subcategories represent?
5. Table 4 and its interpretation for logistic regression were not clear. It involved two or three subcategories of anxiety or depression. It only stated the cut-off point ≥10, which the authors did not explain what it means. Please change the table and interpret the result more comprehensively so the reader may understand.
Author Response
We thank the reviewers for all comments that give us the opportunity to improve the quality of our work. All questions have been carefully analyzed and we have tried to fully respond to what was requested.
“This study determined the levels of anxiety and depression symptoms in Portugal residents during the confinement of the COVID-19 pandemic. The authors described the study well. However, a few comments need attention from the authors to improve this manuscript.”
Thank you very much for your recognition of us.
- The statement for sample size was not clear. In your method, you did mention that the sample participants were 920 (Line 77). There were 9 participants excluded due to missing data, more than 20% (Line 81). However, in your result section, your analysis included 920 participants – has this excluded the 9 participants with missing data? How many participants were used in the analysis, and what was the sample size calculated for this study?
Response: Thanks for pointing this issue out. We apologize for not being clear. We tried to clarify by introducing...
“Sample size was calculated a priori using G × Power 3.1.9.7 software. In the case of depressive and anxiety symptoms, the effect sizes were based on the minimal clinically important differences (MCID) for depressive symptoms (Cohen d = 0.24) [30] and anxiety (Cohen d = 0.29) [31]. In order to use ANOVA with two groups, effect size of Cohen d = 0.24, statistical power (1-B) of 90%, and Alpha of 5%, the minimum sample required was 186 participants. The initial sample consisted of 929 participants. Participants with more than 20% of missing data were excluded (9 participants).” (line 80 to 88)
- Please include in the footnote of Table 3 to differentiate which variable used Student-t and ANOVA test. Why do you need both when ANOVA can cater to two groups and above? Suggest using ANOVA.
Response: We are extremely grateful to the Reviewer for pointing out this issue. We responded according to his suggestion. We put in the footnote : *ANOVA. ** Student-t test However, Student's t Test was used to compare two groups. That was the reason for its use.
- In Table 3, only the health profession was analyzed. What about other professions since the data involved Portuguese residents, not particularly from the health sector only?
Response: Thanks for the comment. In fact, we thought about this aspect during the statistical treatment, however our choice was related to the fact that the group of health professionals had the greatest representation in our sample. With regard to the other professions of the participants, they were disparate, which made a more detailed analysis of this aspect impossible.
Another aspect was due to the fact that we are Nurses, we wanted to assess the differences between different health professionals.
- Categorical variables depression and anxiety in the original scale were 5 subcategories (Line 98) and 4 subcategories (Line 107-108), respectively. Why were they categorized into 3 and 2 subcategories by Visual binning in SPSS (Line 124-125)? Provide the justification and support for such. And what did the new subcategories represent?
- Table 4 and its interpretation for logistic regression were not clear. It involved two or three subcategories of anxiety or depression. It only stated the cut-off point ≥10, which the authors did not explain what it means. Please change the table and interpret the result more comprehensively so the reader may understand.
Response: We decided to answer question 4 and 5 together.
We initially worked with the data from two perspectives. The first was by ordinal logistic regression and the second dichotomous, with cutoff points => 10. In the end we opted for the latter. We are very grateful that you warned us of the error. We sincerely apologize for this error. The correct way is as follows: The odds ratios (ORs) of anxiety and depression symptoms were also calculated by step-by-step multiple logistic regression model from which odds ratios for each explanatory variable with the corresponding 95% confidence interval (CI) and p –value were presented. We used cut-off point ≥10 to GAD-7 and PHQ-9 scales, to create the dichotomous variables. (line 141 to 146).
Reviewer 2 Report
Dear authors,
I am very grateful to have been able to review your article. I hope that my considerations can help you to improve it:
I think that in the abstract starting directly with the objective is not appropriate. I recommend using at least one introductory sentence.
I am struck by the age range of 18 to 77 years. You should indicate why these ages are used and not others.
There is a problem in the sample and that is that, if you only take into account having been detained for 15 days, it is likely that the symptoms of anxiety and depression are previous. Do you ask about previous mental pathology in your survey?
I consider it necessary to include exclusion criteria for the sample.
Also, the PHQ-9 questionnaire will have reliability data that is not included and should be.
The sample is biased, as health professionals have not been in seclusion because they have been working, but their exposure to stressful situations is higher. They cannot be included in the same group as the general population.
It is to be expected that the youngest are the most anxious and depressed.
The reasons why participants attend health consultations are not specified, nor what kind of physical exercise they do. Could they leave their homes to exercise?
Claims are made for the whole Portuguese population, but the sample is not representative of all Portuguese.
Limitations of the study are not included and should appear.
The bibliographical references are partially adequate. There are very interesting studies that have not been included. I indicate two examples:
https://doi.org/10.1016/j.bbi.2020.05.026
https://doi.org/10.3390/ijerph18115732
I wish you luck with the corrections.
Best regards,
The reviewer.
Author Response
Dear Reviewer 2
First of all, we wish to take this opportunity to thank you for the valuable and constructive comments, so that we can see the shortcomings of the paper. These comments are all valuable for improving our paper. All the comments have been carefully revised and highlighted.
I am very grateful to have been able to review your article. I hope that my considerations can help you to improve it:
Thank you very much for your recognition of us.
- “I think that in the abstract starting directly with the objective is not appropriate. I recommend using at least one introductory sentence.”
Response: We are extremely grateful to the Reviewer for pointing out this issue. We responded according to his suggestion…..Background: In previous studies, it was found that the confinement to which the population was subjected during the quarantine of the COVID-19 pandemic increased the risk of anxiety and depression. Please see line 15 to 17.
- I am struck by the age range of 18 to 77 years. You should indicate why these ages are used and not others.
Response: We include people aged 18 and over. The oldest participant was 77 years old. We recognize that this could be a limitation, probably due to the data collection strategy. People who did not have access to email, WhatsApp and internet did not participate. Due to the pandemic situation this was the only way to collect data.
- There is a problem in the sample and that is that, if you only take into account having been detained for 15 days, it is likely that the symptoms of anxiety and depression are previous. Do you ask about previous mental pathology in your survey?
Response: Regarding the quarantine, the average period found was 40.1 days and, in that sense, we apologize, but we do not understand the question raised. However, we are completely available to answer your questions if you are so kind to clarify them for us.
- I consider it necessary to include exclusion criteria for the sample.
Response: We are grateful for the suggestion; however we are informing you that the option of not mentioning the exclusion criteria was due to the fact that we considered as exclusion criteria the non-compliance with the inclusion criteria. However, in view of the suggestion, we add this explanation in line 92 to 93 “We considered as exclusion criteria the non-compliance with the inclusion criteria”
- Also, the PHQ-9 questionnaire will have reliability data that is not included and should be. Response: These data are in the article. please see the line 123.
- The sample is biased, as health professionals have not been in seclusion because they have been working, but their exposure to stressful situations is higher. They cannot be included in the same group as the general population.
Response: We appreciate the comment. Of the 377 health professionals, only 36% continued to work as usual, the rest teleworked or had reduced hours. It may be a limitation of the study. Thank you very much for helping us to improve the manuscript.
- It is to be expected that the youngest are the most anxious and depressed.
Response: We appreciate the comment. Other studies reach the same results. It can probably be due to inexperience of life.
- The reasons why participants attend health consultations are not specified, nor what kind of physical exercise they do. Could they leave their homes to exercise?
Response: Once again, we appreciate the comment. The option not to discriminate the type of medical consultation was due to the fact that, at that time, access to health care was limited, with several types of consultations suspended because they were not considered essential. In this way, we think that discriminating the types of consultation would not bring contributions to the investigation. The type of exercise was not asked. Only if "During the quarantine or social isolation phase, did you do regular physical activity (at least 3 times a week)?"
- Claims are made for the whole Portuguese population, but the sample is not representative of all Portuguese.
Response: Despite the sample comprising 920 people and having participants from all regions of Portugal, wide range of ages and different professional backgrounds, we are aware that the sample is not representative of the entire Portuguese population. We refer to this aspect in the limitations of the study.
- Limitations of the study are not included and should appear.
Response: Some limitations may affect the interpretation of the results of this study, namely the large number of health professionals in the sample, as well as the data collection technique used. Other limitations of the present study are, we present the way which data collection was carried out, as we are aware that not all people residing in Portugal have access to e-mail, internet, or WhatsApp. Finally, another limitation is the fact that 36% of health professionals continue to work, which means that they are not in confinement, which can represent a bias (See line 410 to 416)
- The bibliographical references are partially adequate. There are very interesting studies that have not been included. I indicate two examples: https://doi.org/10.1016/j.bbi.2020.05.026 and https://doi.org/10.3390/ijerph18115732
Response: We are grateful for the suggestion. we integrate article information of the articles. (lines 33to334, and lines 377 to 379)
In conclusion, we have tried our best to improve the manuscript and have made comprehensive changes in this paper. After reading the latest version of our manuscript, we hope you will have a better understanding of this manuscript and look forward to your evaluation.
Best Regards
Round 2
Reviewer 2 Report
Dear authors, I appreciate your effort to adapt to the suggestions made. Still, there are two things to keep in mind:
The exclusion criteria cannot be that the inclusion criteria are not met.
Did they ask about mental pathology prior to the pandemic?
Best regards,
The reviewer.
Author Response
Dear Reviewer 2
We are very grateful for your comments on our work. All these comments are valuable to improve our article. The latest comments have been carefully reviewed and highlighted in yellow.
Dear authors, I appreciate your effort to adapt to the suggestions made.
Thank you very much for your recognition, it was very motivating.
- “Still, there are two things to keep in mind: The exclusion criteria cannot be that the inclusion criteria are not met.
Response: 94 to 97Given the characteristics of the data collection tool, we considered as exclusion criteria low digital competence, the impossibility of accessing the internet or not having social networks and not answering at least 80% of the total questions on the form
- Did they ask about mental pathology prior to the pandemic?
Response: On line 111, it can be seen that in the context of the ad hoc questionnaire on sociodemographic and health variables, participants were asked about pre-existing chronic diseases and about their usual medication. Line 111 to 113 “These data allowed us to assess whether, before the start of the pandemic, individuals already had some type of chronic pathology, namely mental and/or psychiatric.”
We also changed the first sentence of the conclusion: “The research showed that during the first confinement of SARS-COV-2, the mental health of the Portuguese participants in the study was at risk.”
Last Comments: We tried our best to improve the manuscript and made requested changes. We are very grateful for all the contributions you have given us and for the opportunity to improve.
Best Regards